# SUMNAS: Supernet with Unbiased Meta-Features for Neural Architecture Search

**Hyeonmin Ha**
Seoul National University
hyeonmin.ha@snu.ac.kr

**Ji-Hoon Kim**[*]
NAVER AI Lab, NAVER Corporation;
NAVER CLOVA, NAVER Corporation
genesis.kim@navercorp.com

**Semin Park**
Yonsei University
seminpark@google.com

**Byung-Gon Chun**[*]
Seoul National University; FriendliAI
bgchun@snu.ac.kr

## Abstract

One-shot Neural Architecture Search (NAS) usually constructs an over-parameterized network, which we call a supernet, and typically adopts sharing parameters among the sub-models to improve computational efficiency. One-shot NAS often repeatedly samples sub-models from the supernet and trains them to optimize the shared parameters. However, this training strategy suffers from multi-model forgetting. Training a sampled sub-model overrides the previous knowledge learned by the other sub-models, resulting in an unfair performance evaluation between the sub-models. We propose Supernet with Unbiased Meta-Features for Neural Architecture Search (SUMNAS), a supernet learning strategy based on meta-learning to tackle the knowledge forgetting issue. During the training phase, we explicitly address the multi-model forgetting problem and help the supernet learn unbiased meta-features, independent from the sampled sub-models. Once training is over, sub-models can be instantly compared to get the overall ranking or the best sub-model. Our evaluation on the NAS-Bench-201 and MobileNet-based search space demonstrate that SUMNAS shows improved ranking ability and finds architectures whose performance is on par with existing state-of-the-art NAS algorithms.

## 1 Introduction

Recent Neural Architecture Search (NAS) algorithms often train an over-parameterized network called a supernet to obtain supreme sub-models by sharing parameters rapidly. In this process, the shared parameters are generally trained to reach a state where better architecture discovery is possible through comparing the sub-models. Supernet training is usually conducted by sampling one or more sub-models and updating them with their gradients.

Since parameters are shared, learning not to be biased towards a particular sub-model is crucial in training a supernet. However, a parameter update due to sampling a specific sub-model often causes a bias for the sampled sub-model; the supernet forgets the knowledge learned from the previous sampling because of the biased update, and other sub-models share the same parameters can be degraded and underrated. Benyahia et al. (2019) referred to this phenomenon as *multi-model forgetting*, and many current architecture search strategies have been designed without considering this phenomenon.

To mitigate this problem, it is important to make the supernet parameters learn unbiased features that are globally suitable for the sub-models. Therefore, we propose a meta-learning-based approach that enables the supernet to learn unbiased meta-features. We adopt model-agnostic meta-learning (MAML) (Finn et al., 2017; Nichol et al., 2018) during supernet training. MAML is designed to learn the meta-features suitable for multiple tasks, and the trained meta-parameters are then quickly adapted to an unseen task through few-shot learning while reusing the learned meta-features. We

---

[*]Corresponding authors

apply this concept of MAML to supernet training by assuming the multiple tasks in MAML as learning for multiple sub-models in a supernet.

We call the proposed supernet training algorithm Supernet with Unbiased Meta-Features for Neural Architecture Search (SUMNAS), given that it introduces the meta-learning principle to the multi-model forgetting problem. SUMNAS consists of two stages: supernet training and heuristic search with sub-model evaluation. In the supernet training stage, the parameters learn the meta-features for multiple sub-models, which can be considered as MAML's training meta-features for multiple tasks. The meta-parameters obtained from the supernet training can then be directly used without additional training for comparison between sub-models during the evaluation phase.

To the best of our knowledge, this work is the first to utilize the meta-learning capability—learning unbiased meta-features—for accurately ranking sub-models in a supernet. There have been approaches (Shaw et al., 2019; Lian et al., 2020; Wang et al., 2020; Elsken et al., 2020) that adopt meta-learning principles to train a supernet using various tasks and search for the best architecture for an unseen task with a few data instances, which is the few-shot learning variant of NAS. They train the model parameters such that the models are robust to unseen datasets. On the other hand, we take a fundamentally different view of a task. A task usually refers to a dataset, but we show in Section 3 that each sub-model within a search space can be regarded as a separate "task" that the supernet has to adapt to. We therefore apply meta-learning principles to make the model parameters robust to several different sub-models.

We evaluate SUMNAS with qualitative and quantitative experiments on the CIFAR10 and ImageNet datasets. We show that the architecture rankings SUMNAS predicts have a stronger correlation with the accurate rankings as compared to prior NAS algorithms that use a supernet. Besides, we observe better architecture search performance when an existing search methodology is applied to SUMNAS and show that SUMNAS parameters properly learn meta-features by investigating the performances of the sub-models.

## 2 Preliminaries

**One-shot NAS:** In the context of a supernet that shares parameters with its sub-models, the majority of NAS approaches use the supernet as a proxy to indirectly predict the performance of sub-models. With the performance oracle, differentiable techniques (Liu et al., 2018; Cai et al., 2019), reinforcement learning (Pham et al., 2018), and heuristic methods (Guo et al., 2019; Chu et al., 2019) are used to determine the best architecture. Although the weight sharing has significantly improved search speed, it is not easy to provide accurate performance indicators when using a supernet as a proxy for sub-model performance.

One of the most difficult challenges to obtaining an accurate performance oracle is overriding knowledge learned by previously sampled sub-models. Researchers have recently introduced the practice of repeatedly updating parameters through one or more sampled sub-models (Guo et al., 2019; Chu et al., 2019; Li et al., 2020). However, Benyahia et al. (2019) have shown that catastrophic forgetting occurs in the sequential learning of sampled sub-models. Catastrophic forgetting is a phenomenon where a neural network forgets previously learned knowledge when learning new information. In sampling-based supernet training, repetitive sub-model sampling and training keep introducing new information into shared parameters. The knowledge learned with the previously sampled sub-models is forgotten due to training subsequently sampled sub-models without considering former training. Knowledge overriding causes the predicted performance of the architectures to fluctuate depending on the frequency or the sequence of the sampled architectures, which leads to inaccurately predicted rankings. The forgetting problem in the stochastic training of shared supernet parameters is termed *multi-model forgetting*. To alleviate this problem, Benyahia et al. (2019) and Zhang et al. (2020) add a regularizer to prevent weights from deviating too far from the posterior distribution learned from previously trained sub-models.

**Model-agnostic meta-learning:** Model-agnostic meta-learning (MAML) is a process of learning initialization for few-shot learning of an unseen task. Let the $i$th task be $\mathcal{T}_i$ and $\theta_i'$ be suitable to model parameters for $\mathcal{T}_i$. MAML provides an objective function to obtain optimal meta-parameters by minimizing the loss function for each task $\mathcal{T}_i$:

$$\boldsymbol{\theta}_{meta} = \arg\min_{\boldsymbol{\theta}} \; \mathbb{E}_{\mathcal{T}_i \sim p(\mathcal{T})} \left[ \mathcal{L}_{\mathcal{T}_i}(f_{\boldsymbol{\theta}_i^*}) \right] \tag{1}$$

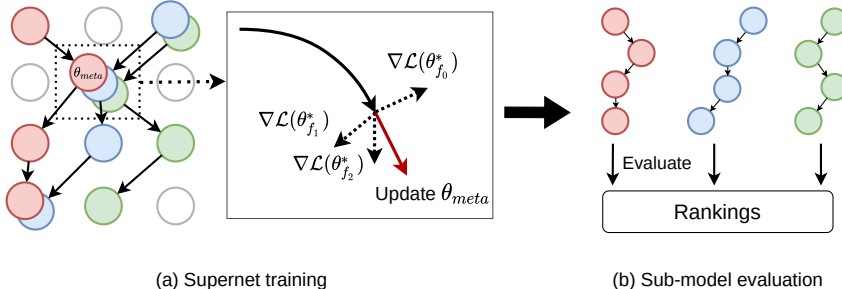

(a) Supernet training           (b) Sub-model evaluation

Figure 1: Two stages of SUMNAS: supernet training and search with evaluation of sub-models. SUMNAS learns meta-features which are globally suitable for the sub-models during supernet training. In the evaluation stage, sub-models are sampled from the trained supernet and are evaluated to decide the outstanding architectures.

where $f$ is the network function consisting of $\theta'$, and $\mathcal{L}$ is the loss function for each task $\mathcal{T}_i$. The $\mathcal{L}$ term in equation 1 allows the $\boldsymbol{\theta}_i^*$ calculated by few-shot learning with initialization $\boldsymbol{\theta}$ to learn a suitable feature for each task $\mathcal{T}_i$, and $\boldsymbol{\theta}_{meta}$ minimizes the expected loss corresponding to the tasks. As a consequence, the learned meta feature is generalized across tasks, allowing for fast learning of new tasks.

Finn et al. (2017) solve the objective function by repeatedly sampling tasks and updating parameters with gradient descents. They also suggest a first-order approximation to avoid the computation of second-order derivatives. Nichol et al. (2018) propose a variation of gradient descent for meta-learning, which is called Reptile. The first-order MAML carries out task optimization using only the last gradient calculated by the inner loop step, ignoring the second-order derivative. However, Reptile uses the average of the gradients calculated over multiple inner loop steps to update the parameters. These averaged values reflect the generalization of the gradients for the data.

## 3 SUPERNET WITH UNBIASED META-FEATURES FOR NEURAL ARCHITECTURE SEARCH

To tackle the multi-model forgetting problem, we suggest a new supernet training strategy based on meta-learning, Supernet with Unbiased Meta-Features for Neural Architecture Search (SUMNAS). As mentioned, learning unbiased features that are suitable for sub-models in the supernet is essential to alleviate the knowledge overriding. We take the idea of MAML, which learns meta-features suitable for multiple tasks, and apply it to supernet training so that SUMNAS learns such unbiased meta-features of sub-models. In this section, we describe our approach and explain how it can mitigate the multi-model forgetting problem. We then describe our supernet training algorithm.

### 3.1 META-FEATURE LEARNING

The approach we introduce to improve robustness against the multi-model forgetting problem is learning unbiased meta-features. SUMNAS has two stages: supernet training and heuristic search with sub-model evaluation, which are similar to existing sampling-based NAS algorithms (Chu et al., 2019; Guo et al., 2019; Li et al., 2020), as presented in Figure 1. First, SUMNAS trains parameters to learn meta-features (Figure 1 [a]), which are not biased to a specific sub-model and are suitable for sub-models, so training them does not overwrite previously trained knowledge. Afterward, the parameters are used to evaluate a specific sub-model during the search (Figure 1 [b]).

Vanilla sampling-based supernet training optimizes strictly shared supernet parameters on sampled sub-models, and it can be expressed as follows:

$$\boldsymbol{\theta}_s = \arg\min_{\boldsymbol{\theta}} \mathbb{E}_{f \sim p(f)}[\mathcal{L}_{\mathcal{T}}(f_{\boldsymbol{\theta}})], \tag{2}$$

where $f$ is a sub-model sampled from a distribution of sub-models $p(f)$, and $\mathcal{T}$ is a given dataset. The parameter $\boldsymbol{\theta}$ is strictly shared among sub-models, and it has to learn model-specific features of all of the sub-models. The training for one or more sampled sub-models does not consider other

ones that have been previously sampled, and simply overrides knowledge acquired from previous training to optimize currently sampled sub-models.

In our approach, similar to previous works, parameters are shared during supernet training. However, the parameters are trained to learn unbiased meta-features. Here, the unbiased meta-features make fair comparison possible by providing each sub-model with an appropriate level of optimized parameters that are not overfitted for a specific sub-model. Our meta-learning approach has been formulated as follows:

$$\boldsymbol{\theta}_s = \arg\min_{\boldsymbol{\theta}} \mathbb{E}_{f\sim p(f)}[\mathcal{L}_{\mathcal{T}}(f_{\boldsymbol{\theta}^*})]$$
$$s.t. \quad \boldsymbol{\theta}^* = \mathcal{A}_f(\mathcal{L}, \mathcal{T}, \boldsymbol{\theta}), \tag{3}$$

where $\mathcal{A}_f$ is an adaptation function optimizing the parameters for the sub-model $f$ with a few inputs. With this optimization, parameters naturally learn meta-features.

This idea is based on MAML, which targets few-shot learning of an unseen task using adaptable parameters. MAML trains parameters to learn meta-features suitable for multiple tasks and uses the meta-parameters to initialize few-shot learning. Training meta-parameters for a weight-shared supernet is similar to training with MAML. We map a task used in meta-learning for few-shot learning to a sub-model in NAS using a supernet, as seen by comparing Equation 1 with Equation 3. In other words, the difference is that MAML minimizes the expectation of the loss across tasks, but SUMNAS minimizes the expectation of the loss across sub-models.

Interestingly, we notice that learning the meta-features in a supernet can be viewed as MAML of each operator in the supernet. We can convert Equation 3 into the following equation:

$$\boldsymbol{\theta}_{meta} = \arg\min_{\boldsymbol{\theta}} \mathbb{E}_{o\sim p(o)}[\mathbb{E}_{f\sim p(f|f^i=o)}[\mathcal{L}_{\mathcal{I}}(f_{\boldsymbol{\theta}^*}^{i+1:} \circ o)]]$$
$$s.t. \quad \mathcal{I} = p(f_{\boldsymbol{\theta}^*}^{0:i-1}(\mathbf{x}), \mathbf{y}|\{\mathbf{x}, \mathbf{y}\} \sim \mathcal{T}) \tag{4}$$
$$\boldsymbol{\theta}^* = \mathcal{A}_f(\mathcal{L}, \mathcal{T}, \boldsymbol{\theta})$$

where $f^{i:j}$ is a composition of sampled operators from $i$-th layer to $j$-th layer. A sampled sub-model $f^{0:i-1}$ feeds an intermediate representation $f^{0:i-1}(\mathbf{x})$ to the operator $o$. There is a distribution of the intermediate representation fed to the operator, and the distribution is determined by a sub-model sampled from $p(f)$. The operator also has a target distribution of outputs to learn determined by the sampled sub-model. In the MAML, examples fed to a meta-learning model also follow a distribution, and this distribution is a task $\mathcal{T}_i$ sampled from $p(\mathcal{T})$. Thus, inputs and targets of both a multi-task meta-learning model and an operator in a supernet follow sampled distributions. We also aim to train an operator that works well on a set of intermediate representations provided by a sampled sub-model, just as MAML learns to work well on a set of examples given by a sampled task. That is, a model and a task in MAML can be mapped to an operator and a distribution of intermediate representations fed to the operator in supernet training. In other words, we can regard the distribution of intermediate representations as to the task for the operator, although the task for the operator is parameterized and optimizable differently from tasks in MAML. This implies that we can use any MAML algorithms without alteration to solve Equation 4. Moreover, it supports the assumption that the success of the meta-learning algorithms for few-shot learning will be transferred to NAS.

However, supernet training and MAML have a fundamental difference. MAML targets to train parameters for an unseen task, but supernet training aims to obtain parameters for pre-defined sub-models, which are known at train time, and participate in the training process. Therefore, after our supernet training, the supernet has knowledge of the sub-models, so additional training, such as fine-tuning for each sub-model is not mandatory. In our empirical results, the additional fitting during evaluation may improve the performance of each sub-model, but the ranking performance stays more or less the same.

## 3.2 SUPERNET TRAINING ALGORITHM

In this section, we present our supernet training algorithm SUMNAS (Algorithm 1). SUMNAS repeatedly samples sub-models and learns meta-features through the sampled sub-models with a MAML-inspired update function. Parameters are updated with the aggregated gradients of multiple sub-models that share the same operators, unlike previous supernet training algorithms that immediately update parameters with the gradients of a single sub-model for an operator.

---

**Algorithm 1** Meta-feature training

---

1: **Input:** training epochs $N$, search layer depth $L$, candidate ops $m$ per layer, training data $D$, steps for adaptation $k$, learning rate $\epsilon_0$ for outer loop and $\epsilon_1$ for inner loop
2: **for** $i \leftarrow 0$ to $N-1$ **do**
3:   **for** $X \in D$ **do**
4:     // Sample $m$ paths
5:     **for** $l \leftarrow 0$ to $L-1$ **do**
6:       $C_{l,:} \leftarrow$ a permutation of the index for the $m$ candidate ops of layer $l$
7:     **end for**
8:     // Compute the Reptile gradients for the sampled paths
9:     **for** $\tilde{i} \leftarrow 0$ to $m-1$ **do**
10:       $\tilde{\boldsymbol{\theta}} \longleftarrow A_{C_{:,i}}^{k}(\mathcal{L}, \boldsymbol{\theta}, X, \epsilon_1)$ // Adaptation
11:       $\boldsymbol{g} \longleftarrow \boldsymbol{g} + \epsilon_0(\boldsymbol{\theta} - \tilde{\boldsymbol{\theta}})$ // Reptile
12:     **end for**
13:     // Update the parameters
14:     $\boldsymbol{\theta} \leftarrow \boldsymbol{\theta} - \boldsymbol{g}$
15:     $\boldsymbol{g} \leftarrow \boldsymbol{0}$
16:   **end for**
17: **end for**

---

**Sampling sub-models** As suggested by FairNAS (Chu et al., 2019), we keep *strict fairness* for fair training among operators, so all of the operators in the supernet are sampled the same number of times and updated simultaneously. To ensure this, SUMNAS sample $m$ sub-models by sampling the candidate operators without replacement for each layer when we have $m$ candidate operators and iterate the sampled sub-models for gradient computation (line 9–12). This sampling process is represented as a permutation of the operators in each layer (line 5–7).

**Meta-learning** The update function to optimize a single sub-model with stochastic gradient descent (SGD) is

$$\boldsymbol{\theta}_f \leftarrow \boldsymbol{\theta}_f - \epsilon \nabla_{\boldsymbol{\theta}_f} \mathcal{L}(\mathbf{y}, f(\mathbf{x}; A_f^k(\mathcal{L}, \boldsymbol{\theta}))), \tag{5}$$

where $\boldsymbol{\theta}$ represents the parameters of the supernet, $\mathcal{L}$ is the loss function, $A_f^k$ is the adaptation function with $k$ steps and a sub-model $f$. We use SGD for the adaptation function. Computing the gradients of the loss requires computing the hessian of the loss, which is too intensive. Several methods can be used to avoid it, and we use the Reptile algorithm (Nichol et al., 2018) because of its simplicity. Reptile uses the difference between the parameters before and after adaptation (line 10–11) instead of the actual gradient. The parameter updates occur after computing and aggregating the gradients of all sampled sub-models (line 13–15).

### 3.3 SEARCH ALGORITHM

Since we have done most of the heavy lifting in the supernet training phase, we can plug in almost any search algorithm for the evaluation phase. Possible search algorithms include, but are not limited to, differentiable architecture search, reinforcement learning, or evolutionary search. For our experiments, we used the simple evolutionary algorithm proposed by Guo et al. (2019).

## 4 EVALUATION

We evaluate SUMNAS on two search spaces — NAS-Bench-201 (Dong & Yang, 2020) on CIFAR-10 (Krizhevsky et al., 2009) and MobileNet blocks on ImageNet (Russakovsky et al., 2015). For both search spaces, we compare sampled architectures' rankings predicted by SUMNAS with their reference rankings obtained from standalone training to evaluate the ranking ability of SUMNAS. We obtain the reference rankings of the sampled architectures for the NAS-Bench-201 search space from the NAS-Bench-201 and those for MobileNet blocks from manual training. We also report the best architectures that SUMNAS found in the search spaces. Furthermore, we analyze the sensitivity of SUMNAS with respect to the number of adaptation steps.

Table 1: Ranking ability of five NAS algorithms and SUMNAS, and top-1 accuracy of architectures the algorithms found on CIFAR-10 under various FLOPs constraints.

| Algorithms | Kendall Tau | Top-1 Acc. under FLOPs contraints | | | |
|---|---|---|---|---|---|
| | | <20M | <50M | <100M | Unlimited |
| DARTS | $0.2987 \pm 0.34$ | 89.36±1.03 | 86.23±4.93 | 86.23±4.93 | 86.23±4.93 |
| GDAS | $0.2231 \pm 0.23$ | 89.96±1.15 | 92.10±0.11 | 92.55±0.33 | **93.26**±0.32 |
| SPOS | $0.8008 \pm 0.02$ | **90.34**±0.03 | 92.33±0.15 | 92.63±0.26 | 92.76±0.02 |
| +WPL | $0.6709 \pm 0.05$ | 89.77±0.46 | 92.35±0.45 | 91.90±0.52 | 92.94±0.78 |
| +NSAS | $0.4769 \pm 0.11$ | 86.85±0.16 | 92.13±0.12 | 92.23±0.10 | 92.23±0.10 |
| FairNAS | $0.7862 \pm 0.01$ | 90.03±0.96 | 92.41±0.15 | 92.34±0.13 | 92.40±0.09 |
| Cream | $0.8100 \pm 0.01$ | 89.91±0.55 | 92.24±0.10 | 92.56±0.41 | 92.83±0.67 |
| **Ours** | **0.8451**±0.01 | 90.30±0.01 | **92.55**±0.22 | **92.93**±0.48 | 93.09±0.12 |

In the experiments, the methodology we use to obtain the predicted performance (*i.e.,* accuracy) of architecture from supernets is the following: we separately update the running statistics of batch normalization layers for the corresponding sub-model and then evaluate the calibrated sub-model. We describe the hyperparameters we used in Appendix D. We also present the analysis on the sensitivity to learning rates for inner and outer loops in Appendix E.

## 4.1 SEARCH SPACE

**NAS-Bench-201** We adopt the NAS-Bench-201 search space to evaluate the ranking performance of SUMNAS on CIFAR-10. NAS-Bench-201 is a benchmark that contains the performance of all of the architectures in its search space on CIFAR-10, CIFAR-100, and ImageNet-16-120 (Chrabaszcz et al., 2017). Each network in the NAS-Bench-201 search space is composed of 15 cells. Each cell has 6 operators and each of them is one of 5 candidate operators: zeroize, identity, 3x3 average pooling, 1x1 convolution, and 3x3 convolution. Cells in a network share the same architecture; therefore, there are $5^6$ possible network architectures in the search space. Details of the search space are described in Appendix A.

**MobileNet blocks** We conduct the experiments on ImageNet using the same search space as ProxylessNAS (Cai et al., 2019) and FairNAS (Chu et al., 2019). The operator in the search space is the bottlenecked and inverted residual block (Sandler et al., 2018) with squeeze and excitation (Hu et al., 2018) and SiLU (swish-1) activation (Ramachandran et al., 2017; Elfwing et al., 2018). This block has been used in various models (Howard et al., 2019; Tan & Le, 2019a; Cai et al., 2019; Chu et al., 2019; Li et al., 2020; Mei et al., 2020). The operators' kernel sizes are either 3, 5, or 7 and can have an expansion ratio of either 3 or 6. In total, there are 6 candidate operators. A candidate network is a sequential execution of the operators (i.e., an operator receives the output from a single previous operator) and has 19 operators in total. Therefore, the search space consists of $6^{19}$ possible candidate network architectures.

## 4.2 RANKING PREDICTION

We compare the ranking ability of the supernet trained by SUMNAS and other NAS algorithms on the NAS-Bench-201 search space and the MobileNet search space. To measure the ranking ability, we use Kendall tau ($-1 \leq \tau \leq 1$) (Kendall, 1938) between the reference rankings from standalone training and the rankings predicted by a NAS algorithm. The details for Kendall tau are described in Appendix B.

**NAS-Bench-201:** To compute Kendall tau, we use the rankings of sampled architectures instead of the entire architecture set defined by the search space. The search space of NAS-Bench-201 contains many architectures with very similar performances, and in many cases, multiple architectures with seemingly different edges could be reduced down to a single architecture due to the topological equilibrium, as shown in Figure 2. Therefore, ranking every single architecture in the search space is likely to produce a noisy list. Furthermore, its orders might change due to other factors, such as which random seed was used during training. Therefore we sample architectures whose performances are different from each other so that the rank correlation could have higher confidence. Specifically, we evenly split the accuracy range of NAS-Bench-201 into 400 intervals and sample

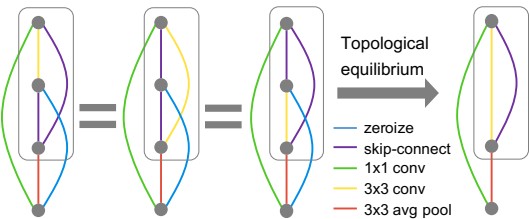

Figure 2: An example of topological equilibrium. Multiple sub-models with different edges could be reduced down to a single architecture. Different combination of skip-connect and 3x3 conv encapsulated in the box produce exactly the same output and thus are all equal to the graph on the far right.

the best one from each interval; 320 architectures are sampled consequently because 80 intervals do not contain any architecture.

The reference accuracies and rankings of the sampled architectures are obtained from the NAS-Bench-201 benchmark. The rankings predicted by DARTS (Liu et al., 2018), ENAS (Pham et al., 2018), and GDAS (Dong & Yang, 2019) come from the checkpoints of supernets trained by the NAS-Bench-201 authors. For SPOS (Guo et al., 2019), FairNAS (Chu et al., 2019), and Cream (Peng et al., 2020), we implement them and train supernets. We also realize WPL (Benyahia et al., 2019) and NSAS (Zhang et al., 2020), which aim to resolve multi-model forgetting problem, on SPOS.

Table 1 shows the ranking ability of the seven NAS algorithms and SUMNAS. We find that SUM-NAS shows more than 4.3% higher Kendall tau value, compared to other state-of-the-art NAS algorithms, such as SPOS, FairNAS, and Cream. Moreover, we note that SUMNAS is superior to other NAS algorithms to resolve multi-model forgetting, such as WPL or NSAS. The outstanding ranking capability of our supernet resulted from the robustness our meta-feature learning achieves against multi-model forgetting. We also check whether other NAS algorithms can earn a gain from more training time since SUMNAS consumes more time to process a single sampled sub-model because of the inner loop (line 10 in Algorithm 1). Specifically, the training time of SUMNAS is longer than FairNAS by a multiple of the adaptation steps because SUMNAS performs forward and backward iterations for each sampled sub-model. For example, when the number of adaptation steps is two, the training time of SUMNAS is twice that of FairNAS. We train supernets with FairNAS as a representative given more training time. FairNAS shows a little improvement, but the gain is still minor comparing that of SUMNAS. The details of the result and analysis are presented in Appendix G.

Table 2: Ranking ability of FairNAS and SUMNAS on MobileNet search space and ImageNet.

| Algorithm | FairNAS | SUMNAS |
|---|---|---|
| Kendall tau | 0.7895 | **0.8526** |

**MobileNet blocks:** We measure the ranking abilities of FairNAS and SUMNAS on the MobileNet search space and ImageNet and present them in Table 2. We sample 20 architectures evenly spaced in MACs from the search space and train them for 100 epochs to get the reference rankings. The Kendall tau of SUMNAS is 7.5% higher than that of FairNAS. SUMNAS more precisely predicts their rankings than FairNAS, as shown in the NAS-Bench-201 search space.

### 4.3 PERFORMANCE OF THE ARCHITECTURE FOUND

We present the best architecture SUMNAS found on the NAS-Bench-201 search space and the Mo-bileNet search space. For the NAS-Bench-201 search space and CIFAR-10, we evaluate all $5^6$ sub-models in the supernets that SUMNAS and other five baselines NAS algorithms trained to discover the best architecture. We then compare the architectures found. For the MobileNet-based search space and ImageNet, where it is computationally infeasible to evaluate all architectures, SUMNAS finds the best architecture through the evolutionary search algorithm described in Section 3.3 and the details for the algorithm and its settings are presented in Appendix C. We present the perfor-

Table 3: Comparison of top-1 accuracies of the model found by SUMNAS, and existing models on ImageNet. Models marked with * are trained with AutoAugment (Cubuk et al., 2018).

| Models | MACs (M) | Params (M) | Top-1 Acc. |
|---|---|---|---|
| MobileNet v3 | 219 | 5.4 | 75.2 |
| MnasNet-A2 | 340 | 4.8 | 75.6 |
| MixNet-M | 360 | 5.0 | 77.0 |
| DNA-B | 406 | 4.9 | 77.5 |
| AtomNAS-C | 363 | 5.9 | **77.6** |
| **SUMNAS-M** | 392 | 5.1 | 77.3 |
| EfficientNet B0* | 390 | 5.3 | 77.1 |
| SE-DARTS+* | 594 | 6.2 | 77.5 |
| FairNAS-A* | 392 | 5.9 | 77.5 |
| **SUMNAS-S*** | 349 | 5.0 | 77.6 |
| **SUMNAS-M*** | 392 | 5.1 | 77.8 |
| **SUMNAS-L*** | 440 | 5.3 | **78.2** |

mances of architectures various NAS algorithms found on their own search space and also report MobileNet-based architectures manually crafted to compare them with ours.

**NAS-Bench-201:** Table 1 shows the top-1 accuracy of the architectures found by each algorithm on CIFAR-10. Without FLOPs constraint, GDAS found the best architecture among the architectures discovered by six algorithms, and our algorithm found the one on par. Considering that GDAS shows low Kendall tau, it is clear that GDAS finds high-performance architectures but fails to properly rank architectures. This becomes a problem for finding the best architectures that meet specific resource constraints, such as memory consumption, MACs, or latency. The performance of architectures found under various MACs constraints and shows this issue. The performances of architectures found by GDAS become worse than those from SUMNAS when such constraints are given, since SUMNAS more accurately rank all the models in the supernet as we mentioned in Section 4.2.

**MobileNet blocks:** The performance of the architecture we found on ImageNet is presented in Table 3 along with the performances of other comparable models (Howard et al., 2019; Tan et al., 2019; Tan & Le, 2019b; Li et al., 2020; Mei et al., 2020; Tan & Le, 2019a; Liang et al., 2019; Chu et al., 2019). All of the models except for MixNet use inverted residual blocks with squeeze and excitation and swish activation as primitive operators. FairNAS and SUMNAS have the same search space, but the main difference between them is our meta-learning principle. Therefore, the performance improvement of SUMNAS should be due to the effect of the meta-learning approach. The model we found also has higher performance than many other state-of-the-art models, such as DNA and SE-DARTS. We also note that some of the other algorithms, such as AtomNAS and DNA, used finer-grained search spaces, with wider choices of the number of blocks and channel sizes of the bottleneck, or incorporates blocks that mix up various kernel sizes. We note that SUMNAS could adopt such search spaces to obtain more superior architectures.

Table 4: The ranking ability of SUMNAS and top-1 accuracies SUMNAS found on NAS-Bench-201 and CIFAR-10 with various adaptation steps. The results are the average of 3 runs.

| Adaptation step | 1 (=FairNAS) | 2 | 3 | 4 |
|---|---|---|---|---|
| Kendall tau | 0.7862±0.02 | 0.8362±0.00 | 0.8361±0.01 | **0.8429**±0.02 |
| Top-1 Acc. | 92.40±0.09 | 92.64±0.30 | 92.92±0.59 | **93.09**±0.12 |

## 4.4 SENSITIVITY ANALYSIS

In this section, we analyze how the performance of our algorithm changes with varying numbers of adaptation steps. We train supernets created with the NAS-Bench-201 search space on CIFAR-10 with combinations of four different adaptation steps {1, 2, 3, 4}. Finally, we measure the Kendall tau between the actual ranking and the ranking predicted by the supernets as in Section 4.2, and the top-1 accuracy of the architectures SUMNAS found.

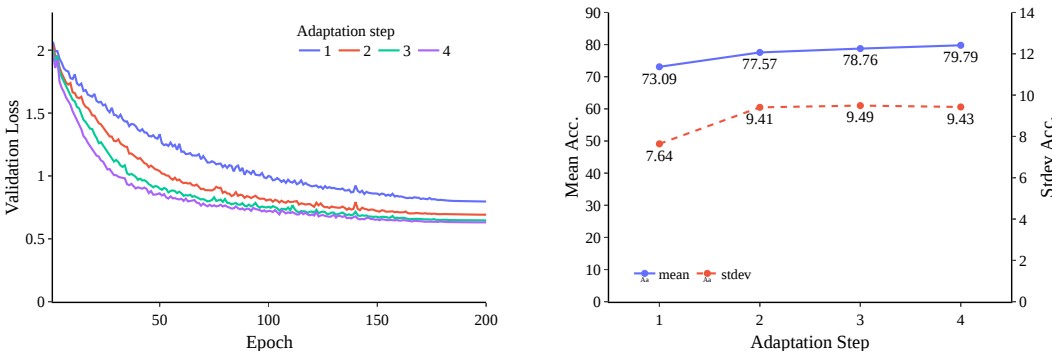

Figure 3: Validation losses of 20 sampled sub-models during SUMNAS supernet training (left), and means and standard deviations of sub-model accuracies with the supernet parameters (right) with various number of adaptation steps on CIFAR-10 and the NAS-Bench-201 search space.

Table 4 shows the results of each configuration. The ranking correlation dramatically increases where the adaptation step increases from one to two, and meta-feature learning is applied. Then the correlation is further improved given more adaptation steps. Our meta-learning principle consistently enhances the ranking prediction regardless of the number of adaptation steps. Top-1 accuracies of the architectures SUMNAS found also show a similar trend to the ranking correlation. We also present the results with the larger number of adaptation steps than four in Appendix H.

## 4.5 Sub-model Performance

Here, we evaluate how meta-feature learning affects the performances of sub-models in a supernet. Since meta-features in SUMNAS cover many sub-models (as opposed to MAML meta-features, which span multiple tasks), the overall sub-model performances using parameters of a supernet trained with SUMNAS are more outstanding than those from a supernet trained without meta-feature learning. We evaluate the performances of sub-models with respect to various adaptation steps to assess the effectiveness of SUMNAS.

The left of Figure 3 shows the averaged validation losses of 20 sampled sub-models during supernet training by SUMNAS, with diverse adaptation steps. SUMNAS with adaptation step 1 (blue lines) does not teach meta-features and is the same as FairNAS. The validation losses dramatically decrease when meta-feature learning is applied, as shown in the results of adaptation step 2 (red). Moreover, the losses further decrease as the number of adaptation steps increases. These observations demonstrate that SUMNAS properly teaches meta-features that sub-models generally utilize.

We also measure the validation accuracies of all sub-models in supernets trained by SUMNAS, with the various number of adaptation steps. The right of Figure 3 presents the mean and standard deviation of the accuracies. The improvement in mean accuracy as the number of adaptation steps increases shows that the overall performance becomes higher with meta-feature learning. This result is consistent with the loss results and also suggests that meta-feature learning is effective. Furthermore, the increased standard deviations indicate that the performances of sub-models are distributed over a wider range. The wider distribution of the performances means performance differences between sub-models become larger and implies a higher resolution of ranking prediction.

## 5 Conclusion

In this paper, we have proposed a new one-shot NAS algorithm which we call SUMNAS, to mitigate the multi-model forgetting problem. In order to achieve this, we took the idea of MAML and applied it to supernet training so that SUMNAS learns unbiased meta-features of the sub-models. By explicitly helping the shared parameters to learn unbiased meta-features, we were able to address the multi-model forgetting problem efficiently. As a result, SUMNAS shows better ranking performance than other state-of-the-art supernet training algorithms, and the performance of the sub-models it finds is on par with hand-crafted models and those discovered by state-of-the-art NAS algorithms.

ACKNOWLEDGEMENTS

This work was supported by SNU-Naver Hyperscale AI Center.

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

## A SEARCH SPACE

**NAS-Bench-201** Architecture in the NAS-Bench-201 search space (Dong & Yang, 2020) used in the proposed method consists of 15 cells and the reduce block is inserted every 5 cells. The architecture has a convolutional layer for the stem before the input layer is connected to the cell, and a global pooling and a fully connected layer at the end, as described in Figure 4 A cell is a densely connected DAG where each node is an intermediate representation and an edge is an operator that transforms the source node and propagates it to the destination node, where the output of multiple incoming edges are combined into one. The reduce block doubles the channel size and halves the height and width of the feature map. Each cell has six edges and four nodes including the input and output nodes. The direction of an edge is from the $i$-th node to $j$-th node where $i < j$. Each edge is one of 5 candidate operators: zeroize, identity, 3x3 average pooling, 1x1 convolution, and 3x3 convolution. All cells of an arbitrary architecture sampled from the search space have the same operator configuration corresponding to each edge. Therefore, there are $|operator\ set|^{\#edge} = 5^6$ candidate architectures in the search space.

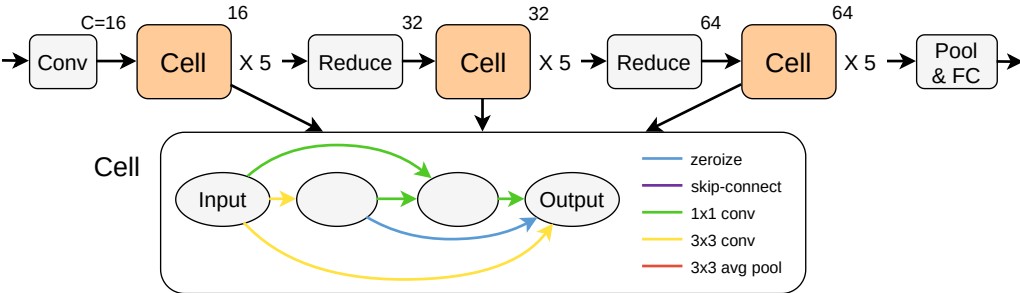

Figure 4: The architecture SUMNAS found on the NAS-Bench-201 search space and CIFAR-10 (Krizhevsky et al., 2009).

**MobileNet Blocks** We use the same MobileNet-based search space (Howard et al., 2019) as ProxylessNAS (Cai et al., 2019) and FairNAS (Chu et al., 2019) in our ImageNet (Russakovsky et al., 2015) experiments for the sake of fair comparison. Each architecture in the search space is a sequential execution of two convolutional layers for the stem, 19 MobileNet blocks, and a convolutional layer and fully connected layer for the head, as depicted in Figure 5.

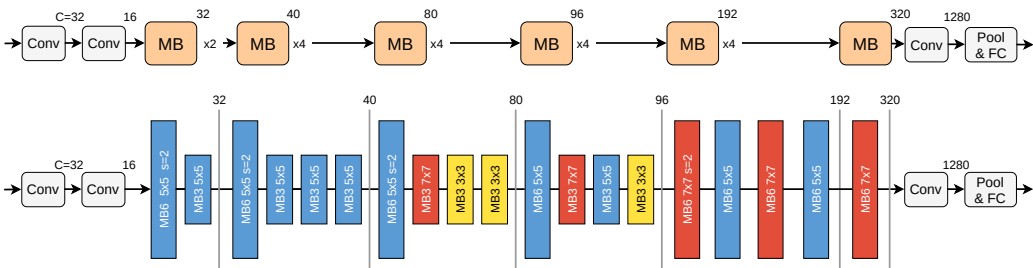

Figure 5: The architecture SUMNAS found on the MobileNet-based search space and ImageNet (Russakovsky et al., 2015). MB$e$ $k \times k$ represents a MobileNet block whose expansion ratio is $e$ and kernel size is $k$. $s$ specifies the stride of the block. The grey line indicates where the channel size is changed and the number over the line is the channel size.

The MobileNet blocks are the bottlenecked and inverted residual blocks (Sandler et al., 2018) that squeeze and excitation (Hu et al., 2018) and SiLU (swish-1) activation (Elfwing et al., 2018; Ramachandran et al., 2017) are mounted. Each architecture consists of blocks with different kernel sizes and expansion ratios. A kernel size of a block is either 3, 5, or 7, and an expansion ratio of a block is either 3 or 6; there are 6 candidate blocks. Architecture in the search space has 19 blocks, and each block is one of the 6 candidate blocks, which means the search space has $6^{19}$ candidate architectures in total.

**Algorithm 2** Evolutionary Search

---

1: **Input:** generations $G$, population $P$, number of offsprings $C$, mutation probability $p$, MACs contraint $M$
2: **Output:** the architecture with the highest validation accuracy that the supernet predict under given MACs constraint.
3: $pop \longleftarrow sample(P)$
4: **for** $i \leftarrow 0$ to $G - 1$ **do**
5:     // Generate offsprings
6:     $m \longleftarrow mutate(p, pop, C/2)$
7:     $c \longleftarrow crossover(pop, C/2)$
8:     $off \longleftarrow m \cup c$
9:     // Evaluate the offsprings and keep the best $P$ samples.
10:     $pop \longleftarrow top\_k(pop \cup off, P)$
11: **end for**
12: **return** $top\_k(pop, 1)$

---

## B    KENDALL TAU

In Section 4.2, we measure the ranking ability with Kendall tau (Kendall, 1938) between standalone validation accuracies from the benchmark and predicted validation accuracies from the supernet. Kendall tau is a measure of rank correlation: the similarity of the rankings of the two variables. Given a set of observations $\{(x_1, y_1), (x_2, y_2), ..., (x_n, y_n)\}$ of the two joint random variable $X$ and $Y$, the measure is the ratio of the discordant pairs of the observations subtracted from the ratio of the concordant pairs. In our experiments, $x_i$ represents a standalone accuracy of architecture from the benchmark and $y_i$ indicates the accuracy of the corresponding architecture that a supernet predicts. The Kendall tau $\tau$ is formally defined as following:

$$
\begin{aligned}
n_c =&|\{((x_i, y_i), (x_j, y_j))|i < j, \\
&((x_i < x_j) \wedge (y_i < y_j)) \\
&\vee ((x_i > x_j) \wedge (y_i > y_j))\}| \\
n_d =&|\{((x_i, y_i), (x_j, y_j))|i < j, \\
&((x_i < x_j) \wedge (y_i > y_j)) \\
&\vee ((x_i < x_j) \wedge (y_i > y_j))\}| \\
\tau =&\frac{n_c - n_d}{\binom{n}{2}}
\end{aligned}
\tag{6}
$$

The number of concordant pairs $n_c$ and the number of discordant pairs $n_d$ is $\binom{n}{2}$ at most; therefore, $-1 < \tau < 1$. If $\tau > 0$, there are more concordant pairs than discordant pairs and this means the two rankings partially (or perfectly when $\tau = 1$) agree. On the other hand, when $\tau < 0$, the two rankings disagree.

## C    SEARCH ALGORITHM

To discover the best architecture on the MobileNet-based search space and ImageNet (Russakovsky et al., 2015), we adopt a simple evolutionary algorithm same as SPOS (Guo et al., 2019). The concrete algorithm is described in Algorithm 2. The search algorithm repeatedly generates offsprings from the population using mutation and crossover, and keep the top $P$ samples among the population and the offsprings for the population of the next iteration. The mutation randomly changes each operator to one of the others with the probability $p$, and the crossover randomly mixes two architectures in the population. The search requires several hyperparameters, and we use generations $G = 30$, population $P = 20$, the number of offsprings $C = 64$, mutation probability $p = 0.1$, and MACs constraint $M = 400(M)$ for the experiment.

Table 5: Search space for the hyperparameters. $LR_{mult}$ is $LR_{inner} \times LR_{outer}$.

|  | CIFAR-10 | ImageNet |
|---|---|---|
| $LR_{mult}$ | {0.01, 0.05, 0.25} | {0.3} |
| $LR_{outer}$ | {0.2, 1, 5, 15} | {0.5, 1} |
| $LR_{outer}$ scheduler | {constant, cosine} | {cosine} |
| $LR_{inner}$ scheduler | {constant, cosine} | {constant} |

Table 6: Hyperparameters we use for the experiments.

|  | CIFAR-10 | ImageNet |
|---|---|---|
| **Outer loop** | | |
| $LR_{outer}$ | 1 | 1 |
| $LR_{outer}$ scheduler | cosine | cosine |
| Optimizer | SGD | SGD |
| Weight decay | 4e-5 | 4e-5 |
| **Inner loop** | | |
| $LR_{inner}$ | 0.05 | 0.3 |
| $LR_{inner}$ scheduler | constant | constant |
| Optimizer | SGD | SGD |
| Opt. momentum | 0.9 | 0.9 |
| Adaptation step | {1,2,3,4} | 2 |

## D    EXPERIMENTAL SETTING

**NAS-Bench-201 search space and CIFAR-10** For the experiments of the NAS-Bench-201 search space (Dong & Yang, 2020) and CIFAR-10 (Krizhevsky et al., 2009), we train the supernets on the entire training set of CIFAR-10. On the test set, the hyperparameters are tuned and the reported Kendall tau and accuracies are measured. We also search for the best architecture on the test set. We tune the hyperparameters where the number of adaptation steps is 4, and the hyperparameter search space is presented in Table 5. We choose the best hyperparameter set which shows the best Kendall tau for the sampled architectures. The searched hyperparameters are listed in Table 6. The tuned hyperparameters are also used in SUMNAS with other numbers of adaptation steps, but the learning rate for the inner loop $LR_{inner}$ is scaled so that actual parameter update sizes keep the same by multiplying $4/adaptation\_step$. SPOS and FairNAS adopt the hyperparameters for the outer loop of SUMNAS, but use $LR_{mult}$ for the learning rate, which is $LR_{outer} \times LR_{inner}$ with scaling as SUMNAS where adaptation step is one.

**MobileNet-based search space and ImageNet** For the experiment of the MobileNet-based search space and ImageNet (Russakovsky et al., 2015), we tune hyperparameters and search the best architecture using a validation set that includes about 50K examples sampled from the training set. Due to the time-consuming task of obtaining the standalone (from scratch) validation accuracies of the sampled architectures to compute Kendall tau, we use the average validation accuracy (from shared parameter) to determine the best performing hyperparameter set. The hyperparameter search space is described in Table 5, and Table 6 presents the selected hyperparameters. We also adopt task batching of MAML and aggregate the gradients of three sub-models for each operator to update its parameters.

## E    PERFORMANCE SENSITIVITY TO THE LEARNING RATES FOR INNER AND OUTER LOOPS

We analyze the sensitivity of SUMNAS performance to the inner and outer learning rates to show that SUMNAS does not demand much extra effort to tune the additional learning rate. We measure

Table 7: Mean and standard deviation of the Kendall tau over the outer learning rates for each LR scale.

| LR scale | 0.01 | 0.05 | 0.25 |
|---|---|---|---|
| mean | 0.8065 | 0.8405 | 0.8372 |
| stdev | 0.0329 | 0.0145 | 0.0164 |

the mean and standard deviation of the Kendall tau over the outer learning rates for each LR scale we suggested in Appendix D. The results are presented in Table 7.

For LR scales of 0.05 and 0.25, the average of the Kendall taus are similar, but at 0.01, the performance is obviously degraded. However, we conducted an ablation study in a sufficiently wided LR scale range, and it can be seen that the degradation is insensitive enough for SUMNAS to the LR scale. The standard deviations show the sensitivity with respect to the relation between the outer and inner learning rates. Standard deviations are usually small, and SUMNAS is robust to the relation of LRs when we take sufficient LR scales, such as 0.05 or 0.25.

## F    DISCOVERED ARCHITECTURES

Figure 4 and Figure 5 visualize the architectures SUMNAS found on the NAS-Bench-201 search space (Dong & Yang, 2020) and the MobilNet-based search space (Howard et al., 2019) respectively.

## G    RANKING ABILITY OF FAIRNAS WITH MORE TRAINING TIME

Table 8: Ranking ability of FairNAS with various training epochs.

| Epochs | 200 | 400 | 600 | 800 |
|---|---|---|---|---|
| Kendall tau | 0.749±0.01 | **0.755**±0.02 | 0.702±0.01 | 0.691±0.01 |

We also check whether FairNAS can earn a gain from more training time, since SUMNAS consumes more time to process a single sampled sub-model because of the inner loop (line 10 in Algorithm 1). Table 8 presents the Kendall tau of FairNAS when more training epochs are given. Note that the learning rate we use here is different from one we present in Appendix D. The supernet for FairNAS in Table 1 is trained for 200 epochs, so the Kendall tau for more than 200 epochs shows the loss or gain from more training time. In 400 epochs, the performance improves for 0.8%. However, the performances degrades in 600 or 800 epochs. We conjecture that the degradation results from cosine learning rate decay, which was used in the FairNAS paper. Increasing the number of epochs might have resulted in longer training time in high learning rates, which might have unstabilized the overall training. Therefore, FairNAS improves slightly with more training time, but the improvement is minor comparing to the gain from SUMNAS.

## H    SUMNAS WITH MORE ADAPTATION STEPS

Table 9: The ranking ability of SUMNAS and top-1 accuracies of architectures that SUMNAS found, given the larger number of adaptation steps than four. The results are averaged over 3 runs.

| Adaptation step | 5 | 6 |
|---|---|---|
| Kendall tau | 0.8339±0.01 | 0.8379±0.00 |
| Top-1 Acc. | 93.05±0.47 | 93.03±0.44 |

Table 9 presents the ranking ability and the performance of the searched architecture on CIFAR-10 and NAS-Bench-201 with the number of adaptation steps above four. The Kendall tau decreases

with the number of adaptation steps when it is larger than four. As the number of adaptation steps increases, each sub-model has to incorporate more data points for a single update, which increases the standard deviation of the gradient that needs to be updated according to the sensitivity of the sub-model. Large standard deviation above a certain level is likely to interfere with learning the unbiased meta feature. Consequently, an excessive adaptation step will have the effect of lowering the ranking ability.

