# OpenReview forum: "SUMNAS: Supernet with Unbiased Meta-Features for Neural Architecture Search"
_ICLR.cc/2022/Conference — ICLR 2022 Poster_

### Official Review · Reviewer_Yfxi · 2021-10-16

**Correctness:** 3
**Technical Novelty And Significance:** 3
**Empirical Novelty And Significance:** 3
**Recommendation:** 8
**Confidence:** 5

**Main Review:**

Strengths
I generally agree with the proposed idea. Naively train the supernet will introduce unfair competition among operators. Instead, the supernet's weights should be adaptive to its subnetworks. Correlation and Accuracy on NAS-Bench-201 and MobileNet space are competitive.

Questions:
1. How sensitive Algo.1 is to the inner and outer learning rate?
2. How does the index permutation work is not well explained (Line 4 in Algo.1). Why is it required? Is this for the purpose of randomness during meta-training? Will there be any gap if no such permutation?
3. Is $g$ aggregating parameter changes ($\theta - \tilde{\theta}$) from all subnetworks (Line 11 in Algo.1)? Will the method work if only part of subnetworks are sampled?
4. I understand that Reptile is efficient since the calculation of Hessian is avoided. I am curious if the authors have ever tried other meta-learning methods, such as the original MAML? If we do not consider computation cost, will the original MAML also work?

**Summary Of The Paper:**

This work targets better ranking performance after supernet training in NAS. The authors leverage meta-learning to make the shared supernet weights adaptive to randomly sampled subnetworks. Experiments are conducted on NAS-Bench-201 and MobileNet space.

**Summary Of The Review:**

I generally like this idea, although seems straightforward, applying meta-learning during training the supernet is a reasonable and correct strategy. The analysis and experiments are also very comprehensive. I vote to accept.

---

> ### Author Response · Authors · 2021-11-19
> **Response to Reviewer Yfxi**
>
> First of all, thank you for your contribution to the conference. We are happy that you agree with the importance of the fair training of the parameter sharing supernet and adaptable meta-learning strategy. The following are responses for your comments.
>
> ## 1. (Sensitivity to the Inner and Outer Learning Rate)
>
> We used grid search: the learning rate scale (LR scale) set is {0.01, 0.05, 0.25} and outer learning rate scale set is {0.2, 1, 5, 15}, where the learning rate scale is defined as inner learning rate $\times$ outer learning rate. Therefore, the inner learning rate can be sufficiently predicted through the combination between learning rate scale set and outer learning rate set. We performed 3 $\times$ 4 ablation studies for adaptation step 4. Details of this experiment are presented in Table 5 in Appendix D.
> To analyze the sensitivity, we measure the mean and standard deviation of the Kendall tau over the outer learning rates for each LR scale.
>
> |LR scale |  0.01 | 0.05 | 0.25 |
> |:------:|----:|------:|------:|
> | mean | 0.8065 | 0.8405 | 0.8372 |
> | stdev | 0.0329 | 0.0145 | 0.0164 |
>
> For LR scales of 0.05 and 0.25, the average of the Kendall taus are similar, but at 0.01, the performance is obviously degraded. However, we conducted an ablation study in a sufficiently wided LR scale range, and it can be seen that the degradation is insensitive enough for SUMNAS to the LR scale. The standard deviations show the sensitivity with respect to the relation between the outer and inner learning rates. Standard deviations are usually small, and SUMNAS is robust to the relation of LRs when we take sufficient LR scales, such as 0.05 or 0.25.
>
> ## (Clarification for SUMNAS Algorithm)
>
> ## Question 2.
>
> The index permutation (Line 4 in Algorithm 1) is the sub-model sampling process. While many one-shot NAS algorithms sample the sub-models completely randomly, we are ensuring strict fairness among the submodels, as introduced in FairNAS. Thus we are sampling sub-models in a way that every operator is sampled exactly once in a training iteration. This is done at the beginning of each iteration, and then the submodels are consumed in the following for loop (Line 9). This sampling process can be implemented with permutation of the operators.
> Suppose a simple random selection process is applied to the sub-model sampling process, not a permutation process. In that case,  there is a possibility that SUMNAS is excessively biased for some operators because the fairness of operator selection is not guaranteed due to the randomness. As a result, the supernet is severely biased to the specific sub-models and will fail to correctly predict the rankings.
>
> ## Question 3.
>
> In Line 11 in Algorithm 1, $g$ aggregates the parameter changes from the subnetworks sampled in Lines 5-6. Each element of $g$ actually aggregates the change from a single subnetwork, because we use FairNAS sampling and the sampled subnetworks do not share any operator.
>
> ## 4. (Using Original MAML)
>
> We have implemented SUMNAS with the original MAML. We will report results as they become available.

---

> > ### Comment · Reviewer_Yfxi · 2021-11-24
> > **Thanks for the response**
> >
> > I read the authors' response and my concerns are addressed. I am especially interested to see if the original MAML works in supernet training.

---

> ### Author Response · Authors · 2021-11-23
> **Answer to Using Original MAML (Reviewer Yfxi)**
>
> Unfortunately, SUMNAS training results using the original MAML are likely to come after the rebuttal deadline has passed. As you expected, the original MAML is computationally intensive and takes more time than expected. This shows that the use of Reptile in our experiments is efficient. Although the original MAML results have not been figured out, we expect it to work well enough even in the original MAML because Reptile is an approximation to avoid the Hessian computation in the original MAML. If this paper gets accepted, we will report the results in a camera-ready version.

---

### Official Review · Reviewer_LNNQ · 2021-10-20

**Correctness:** 3
**Technical Novelty And Significance:** 2
**Empirical Novelty And Significance:** 3
**Recommendation:** 5
**Confidence:** 4

**Main Review:**

This paper utilizes a simple while interesting meta-learning strategy to train the supernet in one-shot NAS, and the experimental results on the NAS-Bench-201 and MobileNet show the effectiveness of the proposed method. However, it seems that the proposed method is a direct application of MAML on one-shot NAS. More importantly, the experimental results can hardly support the claim. For example, although the proposed method achieves a high Kendall Tau value with 0.84 on NAS-Bench-201, we should notice that this search space is very simple, and even without any training, the zero-cost NAS [1] can achieve a similar Kendall Tau value. In addition, paper [2] also consider meta-learning for the neural architecture search.

The marginal improvements in experimental results could hardly convince me that the proposed paradigm can benefit the one-shot NAS. I am confused on the right of Figure 3, where the proposed method SUMNAS achieves 79.79+-9.43 accuracy while Table 3 only achieves 77.3. Please clearly state the difference between the two settings.

[1] Zero-cost Proxies for Lightweight NAS
 [2] M-NAS: Meta Neural Architecture Search

**Summary Of The Paper:**

This paper focuses on improving the predictive ability of supernet in one-shot NAS, which considers a meta-learning strategy to tackle the multi-model forgetting issue. Through utilizing MAML to minimize the expectation of the loss across sub-models, the supernet can learn unbiased meta-features to improve the ranking ability.

**Summary Of The Review:**

The idea is not so surprising and the experiments can not convince me. I tend to reject this paper.

---

> ### Author Response · Authors · 2021-11-15
> **Request of the references**
>
> Thank you for the review. The references seem missing. Could you please provide [1] and [2] so that we can compare SUMNAS to them in the rebuttal?

---

> > ### Comment · Reviewer_LNNQ · 2021-11-21
> > **Reference of [1], [2]**
> >
> > The Reference [1] is Zero-cost Proxies for Lightweight NAS, and [2] is the M-NAS: Meta Neural Architecture Search, which is also cited in this submission. After the clarification, most of my concerns are addressed. I am willing to raise my score.
> >
> > However, the main concern is still about the novelty. This paper permutate several architectures to calculate the mete-gradient, to overcome the multi-model forgetting. This paradigm has been also considered by [3] to relieve forgetting, where [3] also treats different architectures as different tasks. Thus, for me,  this paper is still a direct application of MAML to estimate the gradient for the supernet training. And I also agree that this paper is the first to leverage MAML to estimate the gradient for the supernet training to overcoming multi-model forgetting in NAS.
> >
> > [3] Overcoming multi-model forgetting in one-shot nas with diversity maximization

---

> > > ### Author Response · Authors · 2021-11-21
> > > **Response to Reviewer LNNQ**
> > >
> > > Thank you for your response!
> > >
> > > We understand your concerns about novelty. We believe that the ultimate goal of many parameter-sharing NAS is to overcome multi-model forgetting. However, there can be many different methodologies to achieve this goal, and we reckoned that the meta-learning scheme ingeniously helped overcome this problem. Although we agree that MAML can be viewed as a direct application of gradient estimation, we would like to carefully argue that the sub-model (not a dataset) was not treated as the meta-learning task in the existing MAML-scheme methodologies, is novel. In addition, when our proposed methodology is applied, it shows significant improvement in Kendall tau, a representative indicator of overcoming multi-model forgetting. The reference [3] you mentioned is considered a valuable comment, and we will update it in our paper.

---

> ### Author Response · Authors · 2021-11-19
> **Response to Reviewer LNNQ**
>
> First of all, thank you for your contribution to the conference. Your considerate suggestions improve our paper. The following are responses for your comments.
>
> ## 1. (Novelty)
>
> We would like to clarify that SUMNAS is not a direct application of MAML. Original MAML is a few shot learning technique, where the parameters are trained on various datasets to become robust to an unseen dataset.
> We have reviewed several NAS techniques that adopt meta-learning, and have compared them with SUMNAS in Section 1. The reviewed methods apply MAML to enable the supernet of one-shot NAS to generalize on unseen datasets with only a few examples. Unfortunately, we couldn’t find the reference [2] you mentioned. If [2] is a method we missed, please let us know.
> On the other hand, as we mentioned in Section 1, we introduce a fundamentally different view of NAS, where we regard each sub-model as a task. SUMNAS learns meta-features of sub-models to make the supernet parameters robust to various sub-models in the search space on a single dataset. We describe how this is possible in detail in Section 3.1. To the best of our knowledge, SUMNAS is the first to leverage meta-learning to learn meta-features of sub-models in a supernet and improve ranking ability of the supernet.
>
> ## 2. (Performance)
>
> Reference [1] appears to be missing. We assume that it is Zero-cost Proxies for Lightweight NAS (ICLR 21).
> The zero-cost NAS used Spearman’s rank correlation $\rho$. In the same setting with the zero-cost NAS, the Spearman $\rho$ of the zero-cost NAS, FairNAS and SUMNAS are as follows.
>
> | Algorithm | Zero-cost NAS | FairNAS | SUMNAS |
> |:------|-----:|------:|-------:|
> | Spearman $\rho$ | 0.82 | 0.8453 | 0.8735|
>
>
> As shown in the above table, even if NAS-Bench-201 is simple, SUMNAS outperforms Zero-cost NAS.
>
> ## 3. (Clarification of Figure 3)
>
> Figure 3 shows **the average accuracies of the sampled sub-models on NAS-Bench-201 for CIFAR-10**, and these accuracies are measured on the supernet, **not on stand-alone training models**. The accuracies are not actual performances of the architectures. This figure is intended to verify and illustrate whether SUMNAS has properly trained its meta-features. The improved supernet accuracy suggests that the meta-features trained by SUMNAS are generalized to the sub-models in the supernet and are more robust to multi-model forgetting. You can find more details in Section 4.5.
> On the other hand, Table 3 presents the **standalone training accuracy** of the architecture SUMNAS searched **on ImageNet**. This table shows the actual performance of the architecture.

---

### Official Review · Reviewer_m3L7 · 2021-11-02

**Correctness:** 4
**Technical Novelty And Significance:** 3
**Empirical Novelty And Significance:** 3
**Recommendation:** 6
**Confidence:** 4

**Main Review:**

Strengths:
1.	The paper is well-written and the analysis gives insight.
2.	It addressed an important problem——multi-model forgetting problem by an elegant algorithm based on meta-learning.

Weakness:
1.	The top-1 performance (Table 1 and Table 3) is not impressive although the ranking correlation seems to be improved.
2.	The training time of supernet should also be listed for comparison.
3.	The comparisons are based on less competitive baselines (e.g., I am curious about the performance of KD based method, like BigNAS (https://arxiv.org/abs/2003.11142), Cream (https://arxiv.org/abs/2003.11142), etc.


==========================after rebuttal====================
I appreciate the authors taking the time, attempting to address the comments through more details and new experiments. After reading the authors' response and revisions, I think my concerns are almost addressed. If the authors could present the comparison of training (search) time in the main Table, that will be better. I lean to accept this work.

**Summary Of The Paper:**

This paper addressed the multi-model forgetting problem in supernet training by a supernet learning strategy based on meta-learning. The evaluation on the NAS-Bench-201 and MobileNet-based search space demonstrated an improved ranking correlation and promising performance.

**Summary Of The Review:**

This paper is well-written and the analysis is insightful. However, some related papers are not taken in the comparison, such as BigNAS and Cream.

---

> ### Author Response · Authors · 2021-11-19
> **Response to Reviewer m3L7**
>
> Thank you for your valuable review. Your considerate suggestions improve our paper. To address your concerns, we had a few more experiments for the other method, and we are sorry that the reply is late. The following are responses to your comments.
>
> ## 1. (Performance)
>
> ### 1-1. (Table 1, CIFAR-10 on NAS-Bench-201)
> In NAS-Bench-201 and CIFAR-10, many architectures are densely populated at the high accuracy region, so the absolute improvement may look trifling compared to the enhancement in the ranking ability. However, in rankings, we observed a remarkable improvement with an average rank difference of about 2K among all 16K architectures.
>
> ### 1-1. (Table 3, ImageNet and MAC constraints)
> As we mentioned above to reviewer z8FZ, our relatively small accuracy improvement on ImageNet, despite the huge improvement in Kendall tau, is because of the MACs constraint. Given the current search space, the MACs constraint limits the available sub-model capacity from the supernet, and it can be easily inferred that the highest achievable performance is almost saturated. Therefore, please note that even if there is a significant improvement in Kendall tau, only a small increase in accuracy can be observed.
>
> ## 2. (Training Time)
>
> The training time of SUMNAS is longer than FairNAS by a multiple of the adaptation steps because SUMNAS performs forward and backward iterations for each sampled sub-model. For example, when the number of adaptation steps is two, the training time of SUMNAS is twice that of FairNAS. Please note that even when FairNAS was given longer training time, SUMNAS demonstrated much better ranking ability than FairNAS. The results are presented in Appendix F.
>
> ## 3. (Comparison with KD based methods)
>
> Because of our resource limitations, we provide the results of one of your suggested methods, Cream, which experimented on NAS-Bench-201. We selected Cream because BigNAS only searches over kernel sizes, channel numbers, input resolutions and network depths so it cannot be directly applied to NAS-Bench-201 where we have to search over the operator types (i.e., convolution, pooling, and zero).
> We implemented Cream with FairNAS sampling for a fair comparison. The average Kendall tau and Top-1 accuracy of Cream on NAS-Bench-201 over two runs are 0.8100 and 92.83%, respectively. Thus, Cream shows better ranking ability than FairNAS, but SUMNAS still exhibits superior both of ranking ability and Top-1 accuracy.

---

### Official Review · Reviewer_z8FZ · 2021-11-08

**Correctness:** 3
**Technical Novelty And Significance:** 3
**Empirical Novelty And Significance:** 3
**Recommendation:** 6
**Confidence:** 4

**Main Review:**

Strengths:

The paper is in general well-written and easy to follow. The idea is conceptually simple and part of the results (especially the ones related to Kentall’s tau) are looking promising.

Weaknesses:

Results in Table 4 are supposed to be critical as they aim to verify the effectiveness of the proposed MAML-style supernet training strategy. However, they're not yet fully convincing because some key ablation studies are missing, such as the results without any adaptation step. It would also be interesting to see whether the performance would saturate as we increase the number of adaptation steps further above 4.

I also believe the search space of NAS-Bench-201 is probably too small to make any robust conclusions. While the authors did conduct ImageNet experiments as additional evidence, results there do not appear strong as compared to existing methods. For example, the new method achieved only 0.3% improvement over FairNAS on ImageNet at the same MAdds (Table 3) despite substantially improved Kendall’s tau (Table 2). This makes me wonder whether Kendall’s tau is the right metric to measure the effectiveness of the proposed method. One hypothesis that might explain those observations is that the method indeed improved rank correlations for the vast majority of the "average" architectures but not the calibration for the small number of high-performance ones. An analysis on this direction would be helpful.


**Summary Of The Paper:**

The paper proposed an improved training strategy for oneshot NAS supernetworks. The key idea is to view the training of each subnetwork as a "task", and then to apply an MAML/Reptile-style meta-learning scheme to ensure efficient cross-task adaptivity. Experiments on NAS-Bench-201 and ImageNet show improved calibration between the supernet’s predictions and the architectures' true rankings.

**Summary Of The Review:**

While I'm not fully convinced by all the experimental results (see comments above), this is an overall interesting paper because the method is conceptually simple and the results are looking promising (at least in terms of rank correlations).

---

> ### Author Response · Authors · 2021-11-19
> **Response to Reviewer z8FZ**
>
> We thank you for your valuable review. The following are the responses to your comments. To address your concerns, we had a few more experiments for the ablation studies, and we are sorry that the reply is late.
>
> ## 1. (Performance)
>
> Our relatively small accuracy improvement, despite the huge improvement in Kendall tau, is because of the MACs constraint. Given the current search space, the MACs constraint limits the available sub-model capacity from the supernet, and it can be easily inferred that the highest achievable performance is almost saturated. Therefore, it is not easy to improve the accuracy further.
> At smaller MACs constraint 350M, SUMNAS searched for an architecture of which accuracy is 77.56%. Its accuracy is 0.36 higher than what FairNAS found under the same constraint. For a larger MACs constraint of 450M, SUMNAS is 78.20% accuracy, which is also 0.40 higher than in the 400M of ours. This consistent improvement supports that Kendall tau is still an important metric associated with improving the searched architecture.
>
> ## 2. (Ablation Studies for Adaptation Step)
>
> SUMNAS becomes equivalent to normal training at adaptation step 1. Therefore, the results without adaptation steps should be the results of FairNAS in Table 4.
>
> |Adaptation steps      | 4| 5|        6 |
> |:-------|--------:|--------:|---------:|
> |tau   |  0.8429 | 0.8339 | 0.8379|
> |Top-1 Acc. |    93.09 |   93.05 |   93.03|
>
> This table presents the ranking ability and the top-1 performance of the searched architecture on NAS-Bench-201 for CIFAR-10 dataset with the number of adaptation steps above 3. The results are averaged over 3 runs.
>
> Kendall tau decreases with the number of adaptation steps when it is larger than four. As the number of adaptation steps increases, each sub-model has to incorporate more data points for a single update, which increases the standard deviation of the gradient that needs to be updated according to the sensitivity of the sub-model. Large standard deviation above a certain level is likely to interfere with learning the unbiased meta feature. Consequently, an excessive adaptation step will have the effect of lowering the ranking ability.

---

> > ### Comment · Reviewer_z8FZ · 2021-11-28
> > **Thank you for the reply**
> >
> > My question on adaptation steps has been addressed and I believe including the new results (for adaptation step=4, 5, 6) to Table 4 and adding corresponding discussions would strengthen the paper further.
> >
> > The rebuttal does not change my overall assessment and I'd like to keep my original score.

---

### Author Response · Authors · 2021-11-19
**Major changes of the paper**

We are very excited to have been given the opportunity to revise our manuscript in ICLR2022 OpenReview. We carefully considered those offered by the four reviewers. Herein, we explain how we revised the paper based on those comments and recommendations.

## 1. Comparison with the baseline models (Improved Table 1)

We provide additional baseline model Cream, which Reviewer m3L7 requests, to compare our models under the same search space (NAS-Bench-201). In addition, to provide more explicit comparisons, we implemented Cream with the FairNAS sampling process.

## 2. Clarification of the algorithm (Revised in Section 3.2)

We improve the description for Algorithm 1 in Section 3.2. We clarify the sampling process, represented as index permutation in lines 5–7, and add a clear explanation for the gradient aggregation in line 11 and the parameter update in lines 13–15.

## 3. Training time (Revised in Section 4.2)
At the request of Reviewer m3L7, the training time of SUMNAS was specified compared to FairNAS. Thus, Section 4.2 has been partially revised.

## 4. Clarify Dataset and experiments detail (Improved Figure 3 and revised in section 4.5)
We clearly describe where we conduct the experiments and more explicitly specify the accuracies measured on the sub-models in the supernet, not models trained from scratch. Also, by specifying the dataset used in the experiment, we avoid confusion with the results of other experiments.

## 5. Two additional experiments for the adaptation step (Appendix G)
We measure the performance of the supernet trained by SUMNAS with adaptation step > 4 to check whether the performance is saturated.

---

### Decision · Program_Chairs · 2022-01-20

**Decision:**

Accept (Poster)

**Comment:**

The paper proposes a supernet learning strategy for NAS based on meta-learning to tackle the knowledge forgetting issue. Forgetting happens when training a sampled sub-model to optimize the shared parameters overrides the previous knowledge learned by the other sub-models. The main idea of the paper is to consider training of each subnetwork as a task, and then apply MAML to ensure efficient cross-task adaptivity. While the reviewers found the proposed method mainly an application of the existing meta-learning strategies to one-shot NAS, additional experimental results provided by the authors mostly convinced them about the effectiveness of the proposed method.